# Marine-Derived Ligands of Nicotinic Acetylcholine Receptors in Cancer Research

**DOI:** 10.3390/md23100389

**Published:** 2025-09-30

**Authors:** Igor E. Kasheverov, Irina V. Shelukhina, Yuri N. Utkin, Victor I. Tsetlin

**Affiliations:** Shemyakin-Ovchinnikov Institute of Bioorganic Chemistry of the Russian Academy of Sciences, 117997 Moscow, Russia; shak_ever@yahoo.com (I.E.K.); shelukhina.iv@yandex.ru (I.V.S.); yutkin@yandex.ru (Y.N.U.)

**Keywords:** nicotinic acetylcholine receptors, cancer, neuropathic pain, conotoxins, marine alkaloids, cancer cell markers

## Abstract

Marine sources contain compounds that act on a wide variety of systems, including ligand-gated ion channels. This review will focus on the effectors of nicotinic acetylcholine receptors (nAChRs), for which the diversity of ligands and modulators from marine sources is determined mainly by neurotoxic peptides (α-conotoxins) from mollusks of the *Conus* genus. These are very selective compounds that allow the study of the role of different nAChR subtypes in the cancer cells. They have analgesic or anti-inflammatory activities associated with cholinergic transmission and have shown analgesic effect in case of chemotherapy-induced neuropathic pain. Another class of marine compounds targeting nAChRs for which cytotoxicity for cancer cells was shown is represented by low molecular organic substances found mostly in dinoflagellates and marine sponges. Some of the compounds discussed in this review show promise for developing drugs that suppress cancer growth.

## 1. Introduction 

Cancer is one of the main challenges of medicine today. A prerequisite for successful non-surgical treatment of various types of cancer is understanding the molecular mechanisms of all stages of cancer-cell development, identifying all its participants and, as a result, searching for agents that slow down or even block these stages. By now, a huge amount of material has been accumulated on the conditions leading to the formation of cancer cells; many pathways of their development and the molecular components involved have been identified (see, for example, reviews [1,2,3]).

The cholinergic system including two big families of acetylcholine receptors, muscarinic and nicotinic acetylcholine ones, has been considered as one of the systems involved in the development of some cancer types. Study of the role of metabotropic muscarinic acetylcholine receptors on cancer cells was one of the first to begin and actively continues to this day (see, for example, reviews [4,5,6]); nicotinic acetylcholine receptors (nAChRs) play an equally important role in the development of various cancer types. The discovery of new compounds that control the functions of these receptors with high selectivity is an important task, the solution of which can clarify the role of individual types of receptors in the development of cancer and lay the foundation for the creation of next-generation anticancer drugs. 

nAChRs are pentameric ligand-gated ion channels belonging to the family of Cys-loop receptors that are involved in the neurotransmission throughout the nervous system and in cholinergic signaling in the non-neuronal systems (immune, glial cells, etc.). Historically, nAChRs were divided into muscle and related muscle-type receptors from the fish electric organs (formed by two α1, β1, δ, γ or ε-subunits) and neuronal (non-muscle) receptors formed by different variants of α2−α10 and β2−β4 subunits. “Non-neuronal” nAChRs, which are located outside neurons in other organs, tissues and cells, including the immune system, are built from the same subunits as neuronal nAChRs. Homopentameric receptors can be formed by some α subunits (α7, α8, α9), and heteropentameric ones by various combinations of α and β subunits [7].

Numerous studies have shown altered expression of various nAChR subunits in cancer cells compared to normal cells; for example, overexpression of α5, α7, α9, β4 subunits in breast cancer cells [8,9], or overexpression of α3, α9, α10, β4 subunits in human cervical cancer cell lines [10] were found. Up-regulated expression of α7 nAChR was observed in tissues of gastrointestinal (gastric, colorectal, pancreatic and liver) cancers as compared to normal adjacent tissues (see review [11]), as well as in non-small cell lung cancer [12].

The use of various specific nAChR ligands has made it possible to identify the role of distinct receptor subtypes in carcinogenesis revealing molecular details of this process, including activation of specific signaling pathways (see, for example, the review [13]). Thus, the use of nicotine or its derivatives has demonstrated the important role of distinct nAChR subtypes in the development of various types of cancer, such as α7 nAChR in lung, bladder and colon cancers [14,15,16,17], α9α10 nAChR in lung and breast tumors [17,18,19], as well as in gliomas, glioblastomas and melanomas (see review [20]), and the α3, α5, α7-containing receptors in pancreatic cancers [21]. 

This mini-review considers the use of marine-derived nAChR ligands in cancer studies, as well as works assessing the prospects of such compounds as anticancer agents or potential cancer cell markers. It also provides a brief overview of nAChR-targeting marine compounds that show promise in preventing various types of pain associated with cancer or its treatment. 

## 2. Peptides of Marine Origin as Anticancer Agents Targeting Various Cancer Cell Lines and Inoculated Tumors 

The ability of compounds of various nature to delay the growth of cultured cancer cells, to influence their proliferation or suppress other vital development processes is the first step in assessing the prospects for their further use as anticancer agents. A huge number of such compounds have been described in the literature, including cholinergic ligands of marine origin. Concerning those acting on the nAChR, the leading role since the early 80s of the last century has been played by α-conotoxins, small peptides found in the venom of marine predatory mollusks of the genus *Conus*, which effectively and selectively block various subtypes of these receptors. There is evidence in the literature of the presence of a cytotoxic effect and the ability to induce apoptosis in various cancer cells for whole venoms of some representatives of this genus—*Conus vexillum*, *C. textile*, *C. flavidus* [22,23,24,25]. Not surprisingly, α-conotoxins, some of the most biologically active components of cone snail venoms, have been actively studied in recent years for their cytotoxic properties against various cancer cell lines.

One of the first studies of this kind was the identification of the ability of α-conotoxins MI and ImI (Table 1), antagonists of the muscle and α7 nAChR subtypes, respectively, to inhibit cell proliferation evoked by either nicotine or cytisine in three different human small cell lung carcinoma lines (GLCS, NCI-N592, NCI-H69) [26,27]. Studies on a variety of lung cancer cell lines, where the involvement of the α7 nAChR subtype in the development of this type of cancer is now considered proven (see, for example, the review [28]), are still actively ongoing, but investigations in this direction have also been carried out on other cancer lines. For example, in human monocytic cell line THP-1 derived from an acute monocytic leukemia patient, addition of α-conotoxin ImI resulted in upregulation of pro-inflammatory cytokines TNF-α and IL-8 (but not anti-inflammatory cytokine TGF-β) via α7 nAChR inhibition [29]. 

Conotoxin Cal14.1a, recently discovered in *Conus californicus* venom (Table 1), showed cytotoxic effect on four lung cancer cell lines (H1299, H1437, H1975, H661), namely decreased cell viability mediated through apoptosis by caspase 3 and 7 activation mechanism [30]. Its close analogue, Cal14.1b (Table 1), identified in the same mollusk, was cytotoxic to H1299 cells, but via a caspase-independent mechanism [31].

The involvement of α3/α5/β4-containing nAChRs in lung carcinogenesis was demonstrated in the work on DMS-53 small cell lung carcinoma cells [32]. Nicotine promoted the viability of these cells, while the α3β4 nAChR antagonist α-conotoxin AuIB (Table 1) decreased carcinoma cell viability. Another selective α3β4 nAChR antagonist, α-conotoxin TxID (Table 1), inhibited the growth of two lung cancer cell lines A549 and NCI-H1299 and enhanced the inhibitory effect of adriamycin, a chemotherapy drug used to treat certain types of cancer [33]. TxID also inhibited the growth and proliferation of cervical cancer cell lines SiHa and CaSki [10]. The inhibition of growth and migration of A549 and NCI-H1299 lung cancer cells were also observed when exposed to a fusion protein consisting of α-conotoxin ImI (an α7 nAChR subtype antagonist) and human alpha fetoprotein domain 3 (AFP3), a prototype of anticancer agent [34]. This chimeric product also demonstrated synergistic action with gefitinib, another medication widely used against certain breast, lung and other cancers [34]. 

The regulatory role of heteromeric (α3β2α5) and homomeric (α7) nAChRs in normal and cancer (A549 or H1299) bronchial cells was studied in [35] with the use of gene-silencing method and some nAChR ligands including α-conotoxin MII (Table 1), the antagonist of α3β2 receptor subtype. As a result, a negative role of α3β2α5 nAChR was suggested in the regulation of nicotine signaling through α7 receptors and its impact on cell adhesion, migration/invasiveness and p63 protein expression, which is a critical regulator of epithelial cell differentiation, adhesion and survival.

A number of α-conotoxins (MII, PnIA, RgIA, ArIB[L^11^D^16^]) (Table 1), targeting different subtypes of neuronal nAChRs, have shown the ability to suppress the growth of Ehrlich carcinoma [36,37], increasing antitumor activity of mouse splenocytes [37]. The α-conotoxin ArIB[L^11^D^16^], which is a selective α7 nAChR antagonist, inhibited nicotine-induced cell proliferation of A549 lung cancer cells via the p-Akt pathway [17]. Analogue of α-conotoxin RgIA–RgIA4 (Table 1), potent antagonist of both rat and human α9α10 nAChRs, in a similar way inhibited nicotine-induced cell proliferation of A549 lung cancer cells via the p-Akt and p-ERK pathways [17]. It is interesting that the same set of conotoxins promoted the proliferation of C6 glioma cells and reduced the antiproliferative and cytotoxic effect of lipoxygenase and cyclooxygenase inhibitors on these cells [38]. The literature also contains other data on the stimulating effects of a number of nAChR inhibitors on certain types of cancer cells, in particular, glioblastomas. Thus, α-conotoxins PnIA[L^10^] and RgIA, antagonists of α7 and α9α10 nAChR subtypes, significantly enhanced cell proliferation of the patient-derived glioblastoma cell lines and model glioblastoma U87MG cells in the presence of nAChR agonists under various culturing conditions [39]. 

A representative of another group of conotoxins, αO-conotoxin GeXIVA (Table 1), a potent blocker of α9α10 nAChR identified in the venom of *C. generalis*, has demonstrated analgesic properties in several rat models and showed the ability to inhibit the growth and proliferation of SiHa and CaSki cervical cancer cells [10] and 17 different breast cancer cell lines [9]. In the latter case, a higher mRNA expression of the α9 subunit was observed in all 17 cancer lines compared to the normal breast cell line HS578BST, and that the inhibition of proliferation was due to the interaction of conotoxin with the α9α10 receptor was demonstrated using a knockout of the α9 subunit in the MDA-MB-157 breast cancer line [40]. On the same cells, GeXIVA not only suppressed their proliferation, but also induced apoptosis and decreased the cells abilities to migrate [40]. This conotoxin was no less effective in inhibiting the growth of 4T1 cells of triple-negative breast cancer in vitro and blocked the tumor growth in 4T1 allograft mice at a very low dose of 0.1 nmol per mouse in vivo [41]. It was found that the antitumor mechanism of GeXIVA simultaneously induced caspase-3-dependent apoptosis and blocked proliferation mediated by the downregulation of the AKT-mTOR, STAT3 and NF-κB signaling pathways.

In general, it can be considered that for α3β4, α7, α9α10 nAChRs any peptide antagonist of marine origin may be of interest for the development of an anticancer agent. For example, high anticancer potential was predicted for α-conotoxin RegIIA and especially for its RegIIA[A^11^A^12^] analogue (Table 1), a highly selective antagonist of α3β4 nAChR [42]. Anticancer activity can be expected for the cystine knot folded 42-membered peptide Ms11a-3 (Table 1) recently isolated from the sea anemone *Metridium senile* and characterized as a potent inhibitor of α9α10 (and to a lesser extent α7 and muscle) nAChRs [43].

Another marine source of promising anticancer agents may be sea snake venoms. In any case, several studies on the antitumor activities of whole venoms of sea snakes *Hydrophis spiralis*, *Lapemis curtus*, *Enhydrina schistosa* have been published, but the active components have not been identified [44,45,46].

## 3. Low Molecular Weight Compounds of Marine Origin as Anticancer Agents In Vitro and In Vivo

Biologically active natural compounds from various marine animals are represented not only by compounds of peptide nature, but also by low molecular weight substances, especially alkaloids. Many of them have shown the ability to interact with various nAChRs. Thus, pinnatoxins A-H are produced by dinoflagellates and contaminated shellfish. They are cyclic imine neurotoxins and, as has been previously shown, many of them very effectively bind to orthosteric sites of some nAChR subtypes, in particular muscle, α4β2, α7 ones [47,48]. To date, only pinnatoxin G purified from the marine microalgal species *Vulcanodinium rugosum* (Figure 1) has been studied for its cytotoxicity against cancer cells in vitro. Thus, in the first such study, pinnatoxin G (up to a concentration of 0.46 µM) did not exhibit cytotoxicity for mouse neuroblastoma Neuro2a, human colon adenocarcinoma Caco-2, cervical cancer HeLa cells [49]. However, at a concentration of 5 µM, pinnatoxin G can decrease viability with both cytostatic and cytotoxic effects for six cancer cell lines. More sensitive were HT29 colon cancer, LN18 and U373 glioma cells while MDA-MB-231 breast cancer, PC3 prostate cancer, U87 glioma cells were less sensitive [50].

From marine sponges belonging to the order Haplosclerida, different 3-alkylpyridine and 3-alkylpyridinium compounds were isolated as monomers, cyclic or linear oligomers and high molecular weight polymers which showed a broad spectrum of biological activities including antimicrobial, cytolytic, cytotoxic and antifouling activities (see review [51]). Among them polymeric alkylpyridinium salts (poly-APS), isolated from *Haliclona* (*Rhizoneira*) *sarai* marine sponge, represented a mixture of 2 polymers of 3-octylpyridinium, consisting of 29 and 99 monomeric units. They manifested selective cytotoxicity towards lung cancer cells (A549, Ca-Lu-1, Sk-Lu-1 adenocarcinoma cell lines as well as patients’ non-small cell lung cancer cells) as compared to the normal MRC-5 fibroblast cell line [52]. Single-component synthetic analogues of poly-APS were obtained, some of which demonstrated the ability to interact with nAChRs. Thus, synthetic 3-octylpyridinium salt APS8 (Figure 1) was a potent non-competitive α7 nAChR antagonist which induced apoptosis in non-small cell lung carcinoma (A549, SKMES-1) and HT29 human colon adenocarcinoma cell lines [53,54]. A close synthetic analogue of APS8, APS7, which is a mixture of two cyclic forms (Figure 1), was also an effective nAChR antagonist that inhibits the pro-proliferative and anti-apoptotic effects of nicotine on A549 human lung adenocarcinoma cells [55]. As a potential anticancer drug, it was encapsulated in gelatin-based nanoparticles for targeted delivery to tumors and in this form not only did not lose its inhibitory properties but also caused a stronger decrease in the proliferation of A549 lung cancer cells. It demonstrated a much higher selectivity in cytotoxicity towards cancer cells compared to non-tumorigenic lung BEAS-2B cells, as well as restoring or improving the nicotine-reduced efficacy of chemotherapeutic cisplatin [55]. The same research group also obtained other promising APS analogues for lung cancer therapy, which showed cytotoxicity on A549 cells, blocked the proliferative effects of nicotine and restored the effectiveness of cisplatin via α7 nAChR antagonism [56,57]. 

The data about anticancer activities of above-mentioned marine origin nAChR ligands are summarized in Table 2.

It is worth noting that a large number of marine compounds of non-peptide nature have now been identified, for which effective interaction with various nAChR subtypes, including those involved in carcinogenesis processes, has been demonstrated. In particular, we have identified several similar compounds from sponges, ascidians and nudibranchs that interact with α7 nAChR [58,59]. However, their anticancer potential in relevant cell lines has not yet been investigated. The exception was rhizochalinin (Figure 1), initially isolated from the marine sponge *Rhizochalina incrustata* and showing micromolar affinity for muscle-type and neuronal α7 nAChRs [58], which significantly reduced viability of PC-3, DU145, LNCaP, 22Rv1, VCaP human prostate cancer cell lines in vitro and PC-3 and 22Rv1 tumor in vivo inducing caspase-dependent apoptosis. However, voltage-gated potassium channels have been identified as a main molecular target of rhizochalinin [60]. 

On the other hand, there are still more marine compounds that have shown significant cytotoxic or antiproliferative effects on various cancer cell lines that are not related to the action on the nAChRs or have not been studied in this respect. An example illustrating this fact can be found in review [61], where data on the alkaloids of sponges and ascidians with structures containing dibrominated indolic systems were discussed. Among them, many natural compounds or their synthetic analogues (dibromotryptamines, dragmacidin, brominated-β-carbolines, meridianins, aplicyanins and others) are considered as promising anticancer agents and targets of action have even been identified for some of them but not as nAChRs. Interestingly, a mono-brominated indole compound (6-bromohypaphorine) found in the nudibranch *Hermissenda crassicornis* was tested for its ability to interact with nAChR and was found to be an agonist of α7 nAChR with micromolar affinity [59]. Its analogues with increased affinity for this receptor were synthesized; two of them, 6ID and 6ND (Figure 1), showed high anti-inflammatory activity on macrophages, as well as noticeable effects in a number of anti-pain and anti-inflammatory tests in rodent models in vivo [62]. The question of the anticancer activity of these hypaphorine analogues still remains open. 

## 4. Marine Compounds as Potential Analgesic Agents for Pain Relief in Cancer Therapy 

The processes of carcinogenesis, as well as the treatment of cancer, are associated with the occurrence of pain. A huge number of studies aimed at finding or development of analgesic agents capable of suppressing this pain have been carried out. Among the large number of potential analgesics described, there are also compounds of marine origin. A striking example here is Ziconotide (a synthetic version of the ω-conotoxin MVIIA, which blocks N-type calcium channels), which was introduced into medical practice more than 10 years ago and is used to relieve chronic pain in the treatment of cancer patients (see, for example, review [63]). 

Ligands of nAChR (including those of marine origin) have also been actively studied in the last two decades as possible analgesics, especially in the processes of inflammation, peripheral neuropathy and carcinogenesis. By now, it has become clear that the most significant role in the first two processes is played by α7 and α9α10 nAChRs (see, for example, reviews [64,65]) and, accordingly, selective ligands of these two nAChR subtypes are the most promising objects for study as potential analgesics. A separate direction here is the search for compounds for suppressing chemotherapy-induced peripheral neuropathy (CIPN) that occurs commonly during cancer management, which often limits the dosage of medication or even leads to its withdrawal. Examples of such anticancer medications include the most widely used oxaliplatin and paclitaxel. 

Some of the first positive results here were obtained for compounds inhibiting the α9α10 nAChR subtype. Thus, one of the isomers of conotoxin GeXIVA can relieve and reverse oxaliplatin-induced mechanical and cold allodynia after single and repeated intramuscular injections in rats [66]. A similar analgesic effect of this isomer was also observed in a mice model, and its ability to alter expression of genes associated with immune-related pathways represented by the cytokine–cytokine receptor interaction pathway was demonstrated [67]. 

One more example is α-conotoxin RgIA, another α9α10 nAChR antagonist, which has been shown in many studies to be able to prevent chemotherapy-induced neuropathic pain. Therefore, the administration of RgIA [68] reduced the oxaliplatin-dependent hypersensitivity to mechanical and thermal noxious and non-noxious stimuli and significantly prevented morphological modifications of L4–L5 dorsal root ganglia. Similar positive effects on the long-term oxaliplatin-induced peripheral neuropathy in mice were confirmed for α-conotoxin RgIA in tests of cold allodynia, hot plate, Von Frey and grip strength analysis. The same effects were discovered for the first time for the octaoligoarginine [69].

RgIA4, a designed analogue of α-conotoxin RgIA (Table 1), which, unlike the original toxin, exhibits nanomolar affinity not only for rat but also for human α9α10 nAChR, effectively prevented oxaliplatin-induced neuropathic pain in both rats and mice [70,71] and paclitaxel-induced neuropathic pain in rats [72]. Several other, more potent and selective for human α9α10 nAChR α-conotoxin RgIA analogues have been obtained. For example, these are RgIA-5524 [73] and RgIA-5474 [74], containing a non-natural cysteine bond (methylene thioacetal or penicillamine, respectively), as well as some non-natural amino acid residues, that resulted in significantly reduced degradation of the peptides in human serum. Both showed exceptional selectivity and affinity (IC_50_ = 0.9 and 0.05 nM, respectively) towards human α9α10 nAChR and effectively reversed CIPN (oxaliplatin-induced cold allodynia) in rodent model.

**Table 2 marinedrugs-23-00389-t002:** Effects of marine origin ligands of nAChR on cancer cell line and carcinoma.

Compound	Main Target	Cancer Cell Line/Carcinoma	Observed Effects	Ref.
α-conotoxins MI and ImI	muscle-type and α7 nAChRs	GLCS, NCI-N592, NCI-H69 small cell lung carcinoma	inhibited serotonin release and cell proliferation evoked by nicotine or cytisine	[26,27]
α-conotoxin ImI	α7 nAChR	THP-1 monocytic leukemia line	increased release of TNF-α and IL-8	[29]
conotoxins Cal14.1a and Cal14.1b	nAChRs	H1299, H1437, H1975, H661 lung cancer cell lines	decreased cell viability	[30,31]
α-conotoxin AuIB	α3β4 nAChR	DMS-53 small cell lung carcinoma cells	inhibited cell viability	[32]
α-conotoxin MII	α3/α6β2-containing nAChRs	A549 and H1299 lung cancer cells	increased cell migration	[35]
α-conotoxin MII	α3/α6β2-containing nAChRs	Erlich carcinoma	increased the cytotoxic effect of indomethacin; inhibited cell growth; increased mouse survival in vivo	[36]
α-conotoxin PnIA	α7 nAChR	Erlich carcinoma	increased the cytotoxic effect of baicalein; inhibited growth cells; increased mouse survival in vivo	[36]
α-conotoxin ArIB[L^11^D^16^]	α7 nAChRs	A549 lung cancer cells	inhibited cell proliferation	[17]
α-conotoxin RgIA4	α9α10 nAChR	A549 lung cancer cells	inhibited cell proliferation	[17]
α-conotoxins PnIA, ArIB[L^11^D^16^], RgIA	α3β2, α7, α9α10 nAChRs	C6 glioma cells	enhanced cell proliferation, decreased the cytotoxic effect of baicalein	[38]
α-conotoxin TxID	α3β4 nAChR	A549 and NCI-H1299 lung cancer cells	inhibited cell growth; enhanced the inhibitory effect of adriamycin	[33]
α-conotoxin TxID	α3β4 nAChR	SiHa and CaSki cervical cancer cells	inhibited cell proliferation	[10]
αO-conotoxin GeXIVA	α9α10 nAChR	SiHa and CaSki cervical cancer cells	inhibited cell proliferation	[10]
αO-conotoxin GeXIVA	α9α10 nAChR	17 different breast cancer lines	inhibited cell proliferation	[9]
αO-conotoxin GeXIVA	α9α10 nAChR	MDA-MB-157 breast cancer line	inhibited cell proliferation; apoptosis; decreased ability of cell migration	[40]
αO-conotoxin GeXIVA	α9α10 nAChR	4T1 triple-negative breast cancer cells	suppressed the cell growth	[41]
α-conotoxin ImI-AFP3	α7 nAChR	A549 and NCI-H1299 lung cancer cells	inhibited cell growth and migration; enhanced the inhibitory effect of gefitinib	[34]
α-conotoxin PnIA[L^10^]	α7 nAChR	patient-derived glioblastoma and U87MG	enhanced cell proliferation	[39]
α-conotoxin RgIA	α9α10 nAChR	patient-derived glioblastoma and U87MG	enhanced cell proliferation	[39]
conotoxin lt14a	nAChRs	HepG2 liver carcinoma cells	decreased cell viability	[75]
pinnatoxin G	muscle-type, α4β2, α7 nAChRs	LN18, U87, U373 glioma cells, MDA-MB-231 breast cancer, PC3 prostate cancer, HT29 colon cancer cells	decreased cell viability	[50]
APS8	α7 nAChR	A549, SKMES-1 lung carcinoma and HT29 human colon adenocarcinoma cells	inhibited growth of tumor cells, prevented regrowth of tumors, reduced the adverse anti-apoptotic effects of nicotine, impaired the cell viability, induced apoptosis	[53,54]
APS7	α7 nAChR	A549 lung carcinoma cells	inhibited nicotine-induced Ca^2+^ influx and proliferation, inhibited nicotine-suppressed apoptosis; restored or improved the nicotine-reduced efficacy of chemotherapeutic cisplatin	[55]

Potential anticancer chemotherapeutics are also suggested on the basis of other conotoxins with pronounced analgesic activity (see, for example, review [13]). Among them are conotoxin lt14a and its analogue (Table 1) that exhibited potent long-lasting analgesia in the hot plate pain model in mice but a slight effect (10–20%) in reduction of the viability of liver carcinoma HepG2 cells (Table 2) [75]. Another example is Mr1.1 [S4Dap], an analogue of α-conotoxin Mr1.1 (Table 1), a potent antagonist of human α9α10 nAChR (IC_50_ = 4.0 nM), which showed analgesic activity in the rat chronic constriction injury pain model (paw withdrawal threshold) [76]. 

Outside the above-considered nAChR subtypes, there are practically no data about identified analgesic properties of marine origin compounds. As an exception, it is worth mentioning the work where analgesic activities were revealed in mouse-twisting tests for four recombinant toxins SNT-1, SNT-2, SNT-3, SNT-4 from sea snake *Lapemis hardwickii* venom with inhibition efficiencies of 32.3%, 34.6%, 25.5%, 55.9%, respectively, at doses of 1/4 LD_50_ [77]. All of them are short chain neurotoxins of 60 amino acid residues and are thought to act on muscle-type nAChRs.

## 5. Marine Compounds as Potential Cancer Cell Markers 

A separate practical direction in cancer research is the development of markers for the detection of cancer cells based on natural or synthetic compounds. Several studies have been carried out on the preparation of fluorescently labeled ligands for distinct nAChR subtypes expressed in some cancer cell lines. Thus, three α-conotoxins TP-2212-59, TxID[A^9^], RegIIA, selective ligands of α3β4 nAChR, bearing FITC-moiety at the N-terminus via a β-alanine link (Table 1), were tested for the ability to specifically bind to nAChR on the neuroblastoma IMR-32 cell line using flow cytometry [78]. However, none of them adequately stained neuroblastoma nAChRs. More successful were the studies with α-conotoxin TxID labeled at the N-terminus with a fluorescent dye 5-TAMRA-SE, which maintained the same order of potency as the native conotoxin for the α3β4 nAChR subtype and labeled this receptor in the RAW264.7 cells [79].

Visualization of other nAChRs with the aid of fluorescent probes based on selective conotoxins was also successful. Thus, three different fluorescent labels (6-TAMRA-SE, Cy3 NHS ester, BODIPY-FL NHS ester) were used to obtain fluorescent derivatives of the α-conotoxin analogue LvIB—LvIB[G^1^, ∆R^14^], a selective antagonist of α7 nAChR, which retained the ability to effectively interact with this receptor subtype and successfully visualized it on rat brain slices [80]. In a similar manner, the same set of fluorescent analogues of another α-conotoxin LvIC[G^1^, ∆Q^14^], that selectively and potently blocks α6/α3β4 nAChR, was obtained and the BODIPY derivative proved to be the most promising [81]. Even earlier, fluorescent analogues of α-conotoxins MII (antagonist of α6- and α3-containing nAChRs), RgIA (selective antagonist of α9α10 nAChR), LtIA (selective antagonist of α3β2 nAChR) were obtained and to a significant extent retained their affinity for the corresponding targets [82,83,84]. Since all of these targets are also present on cancer cells, these fluorescent conotoxins can be used to visualize the respective nAChRs. 

An unusual approach of using conotoxins as a delivery carrier for anticancer agents to target the respective nAChRs was applied in the works [85,86]. α7 nAChR antagonist—α-conotoxin ImI was conjugated to phospholipid micelles with paclitaxel and docetaxel, two chemotherapeutics used for the treatment of lung and breast cancers, respectively. Both constructs not only showed greater cytotoxic and apoptotic effects in human MCF-7 breast cancer cells and A549 lung cancer cells, respectively, as compared to ImI-free micelles, but they were also more effective in delivering both therapeutics in tumor-bearing mice.

## 6. Conclusions and Future Directions

Thus, the presented results show that low molecular weight compounds isolated from marine sources, as well as conotoxins and their derivatives, make it possible to reliably identify certain subtypes of nAChRs in cancer cells. In addition, marine compounds targeting nAChRs are promising candidates for suppressing the growth of cancer cells and tumors both alone and in combination with known chemotherapeutic drugs. Some of the described ligands have shown analgesic effects in case of chemotherapy-induced neuropathic pain.

The presented results also allow us to make general assumptions about future directions of research in this area. New compounds of marine origin with anticancer or analgesic properties and targeting various nAChR subtypes, including substances isolated from poorly studied species of marine animals, will be identified. Work will be intensified to introduce such promising compounds into preclinical studies, for which non-toxic and stable in vivo analogs of these compounds will be designed and synthesized. 

## Figures and Tables

**Figure 1 marinedrugs-23-00389-f001:**
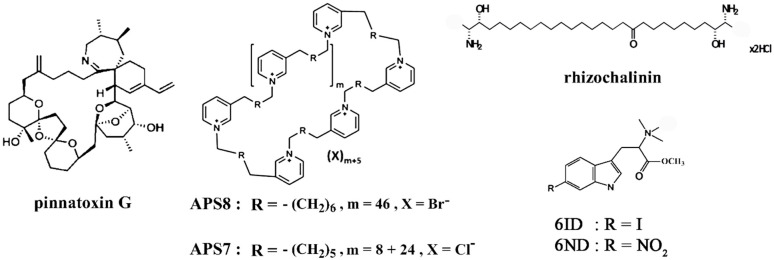
Structure of non-peptide compounds of marine origin targeting nAChR with possible anticancer potential.

**Table 1 marinedrugs-23-00389-t001:** Amino acid sequences of peptide compounds acting on different nAChR subtypes that were used in studies on cancer cells. The disulfide bonds in globular isomers of conotoxins are formed according to the scheme C1–C3, C2–C4.

Name	Organism	Sequence
MI	*Conus magus*	GRCCHPACGKNYSC *
ImI	*Conus imperialis*	GCCSDPRCAWRC *
Cal14.1a	*Conus californicus*	GDCPPWCVGARCRAEKC
Cal14.1b	*Conus californicus*	GDCPPWCVGARCRAGKC
AuIB	*Conus aulicus*	GCCSYPPCFATNPDC *
TxID	*Conus textile*	GCCSHPVCSAMSPIC *
FITC-β-TxID[A^9^]	*Conus textile*	*Fitc*-(Bal)GCCSHPVCAAMSPIC
FITC-β-TP-2212-59	*Conus bullatus*	*Fitc*-(Bal)GCCSHPBCFBZYC
FITC-β-RegIIA	*Conus regius*	*Fitc*-(Bal)GCCSHPACNVNNPHIC
MII	*Conus magus*	GCCSNPVCHLEHSNLC *
PnIA	*Conus pennaceus*	GCCSLPPCAANNPDYC *
PnIA[L^10^]	*Conus pennaceus*	GCCSLPPCALANNPDYC *
RgIA	*Conus regius*	GCCSDPRCRYRCR
RgIA4	*Conus regius*	GCCTDPRC(Cyt)(Yio)QCY
ArIB[L^11^D^16^]	*Conus arenatus*	DECCSNPACRLNNPHDCRRR
GeXIVA	*Conus generalis*	TCRSSGRYCRSPYDRRRRYCRRITDACV *
lt14a	*Conus litteratus*	MCPPLCKPSCTNC *
lt14a[A^7^]	*Conus litteratus*	MCPPLCAPSCTNC *
Mr1.1[Dap^4^]	*Conus marmoreus*	GCC(Dap)HPACSVNNPDIC *
RegIIA	*Conus regius*	GCCSHPACNVNNPHIC *
RegIIA[A^11^A^12^]	*Conus regius*	GCCSHPACNVAAPHIC *
^1^ Ms11a-3	*Metridium senile*	GCKKLNSYCTRQHRECCHGLVCRRPDYGIGRGILWKCTRARK

^1^ For the inhibitor cystine knot folded Ms11a-3 the disulfides are C1-C4, C2-C5, C3-C6. * Amidated C-terminus, FITC (*Fitc*)—fluorescein isothiocyanate, Bal—β-alanine, B—2-amibobutyric acid, Z—norvaline, Cyt—citrulline, Yio—3-iodo-tyrosine, Dap—L-2,3-diaminopropionic acid.

## Data Availability

Not applicable.

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
