# Peer review of "Marine-Derived Ligands of Nicotinic Acetylcholine Receptors in Cancer Research"

_marinedrugs, 2025, doi:10.3390/md23100389_

Round 1
Reviewer 1 Report
Comments and Suggestions for Authors
This is a timely and concise review on diverse ligands for nicotinic acetylcholine receptors (nAChR) subtypes involved in cancer.
The authors introduce their work in a natural way: from generalities on cancer, to the cholinergic system, the structure and subtypes of nAChR, the expression of particular nAChRs subtypes in diverse cancer cells and tissues, the application of ligands for nAChR to study the role of these receptors in the development of distinct types of cancer, and the application of both peptidic and non-peptidic ligands from marine sources as anticancer agents or markers and as molecules to treat pain associated with cancer itself but also with some cancer treatments.
Then, the authors present toxins from cone snails and their derivatives, and a peptide from a sea anemone that have been used in studies on cancer cells (Table 1) and describe in detail their activities (effects) on diverse cell lines and tumors.
The next section focused on low molecular-weight molecules from dinoflagelltes and sponges. Again, the authors present their structures (Figure 1) and recount their effects in detail. Table 2 includes both peptide ligands and non-peptidic ligands.
Following the same style, the last two sections include molecules for relieving pain in both cancers themselves and their treatments, and for employing them as probable markers for cancer cells.
Finally, in the final section, the authors emphasize that diverse molecules from marine organisms allow the identification of particular subtypes of nAChR in diverse cancer cells and that these compounds may suppress the growth of tumors by themselves or combined with drugs currently used in the chemotherapy of cancer and that some have been shown to have analgesic effect during these treatments. They also mention several broad expectations on the future directions in this research field.
Minor issues:
Lines 129 and 286: please, change the Russian character to "and".
Line 239: please, correct the font size.
Table 2: For uniformity, please change "I" to "i" (twice).

Author Response
Summary
Thank you very much for taking the time to review this manuscript and for rating it so highly. We have agreed with all your comments and have made appropriate corrections.
Comment 1: Lines 129 and 286: please, change the Russian character to "and".
Response 1: Thank you for pointing this out. It was corrected.
Comment 2: Line 239: please, correct the font size.
Response 2: Thank you for pointing this out. It was corrected.
Comment 3: Table 2: For uniformity, please change "I" to "i" (twice).
Response 3: Thank you for pointing this out. It was corrected.
Reviewer 2 Report
Comments and Suggestions for Authors
Dear Authors,
The manuscript "Marine origin ligands of nicotinic acetylcholine receptors in cancer research" is a well written, comprehensive review of the current knowledge of the role of nicotinic acetyl;choline receptors in cancer development and potential therapeutic properties of ligands, specifically those having a marine origin, targeting this receptor family. This work will provide a helpful reference for anyone studying or wanting to initiate such studies on this family of receptors and their link to cancer development and therapeutics. I only have very minor grammatical comments and suggestions.
1) For the title, instead of "Marine Origin Ligands of Nicotinic...", I suggest using "Marine-derived ligands of Nicotinic acetylcholine..."
2) In the abstract, line 13 "These are very selective compounds that allow to study...", correct the grammar to "These are very selective compounds that allow study of the role of different nAChR..."
3) Line 153, "...apoptosis and decreased the cell abilities of migration." correct the grammar to "...apoptosis and decreased the cells abilities to migrate."
4) Line 286, there is a strange symbol between RgIA-5524 and RgIA-5474. Please correct.
5) Line2 337-339, correct the following sentence "Both constructs showed greater cytotoxic and apoptotic effects as compared with ImI-free micelles in human MCF-7 breast cancer cells and A549 lung cancer cells, respectively, as well as they were more effective in delivering both therapeutics in tumor-bearing mice" to "Both constructs not only showed greater cytotoxic and apoptotic effects in human MCF-7 breast cancer cells and A549 lung cancer cells, respectively, as compared to ImI-free micelles, they were more also more effective in delivering both therapeutics in tumor-bearing mice."
Author Response
Summary
Thank you very much for taking the time to review this manuscript and for rating it so highly. We have agreed with all your comments, have changed the title and have made appropriate corrections.
Comment 1: For the title, instead of "Marine Origin Ligands of Nicotinic...", I suggest using "Marine-derived ligands of Nicotinic acetylcholine...".
Response 1: We agree with your suggestion. Therefore, we have changed the manuscript title.
Comment 2: In the abstract, line 13 "These are very selective compounds that allow to study...", correct the grammar to "These are very selective compounds that allow study of the role of different nAChR..."
Response 2: Thank you for pointing this out. It was corrected.
Comment 3: Line 153, "...apoptosis and decreased the cell abilities of migration." correct the grammar to "...apoptosis and decreased the cells abilities to migrate."
Response 3: Thank you for pointing this out. It was corrected.
Comment 4: Line 286, there is a strange symbol between RgIA-5524 and RgIA-5474. Please correct.
Response 4: Thank you for pointing this out. A strange symbol has been replaced with "and".
Comment 5: Line2 337-339, correct the following sentence "Both constructs showed greater cytotoxic and apoptotic effects as compared with ImI-free micelles in human MCF-7 breast cancer cells and A549 lung cancer cells, respectively, as well as they were more effective in delivering both therapeutics in tumor-bearing mice" to "Both constructs not only showed greater cytotoxic and apoptotic effects in human MCF-7 breast cancer cells and A549 lung cancer cells, respectively, as compared to ImI-free micelles, they were more also more effective in delivering both therapeutics in tumor-bearing mice."
Response 5: Thank you for pointing this out. The sentence was corrected.